# Identification and Distribution of Sterols, Bile Acids, and Acylcarnitines by LC–MS/MS in Humans, Mice, and Pigs—A Qualitative Analysis

**DOI:** 10.3390/metabo12010049

**Published:** 2022-01-07

**Authors:** Ambrin Farizah Babu, Ville Mikael Koistinen, Soile Turunen, Gloria Solano-Aguilar, Joseph F. Urban, Iman Zarei, Kati Hanhineva

**Affiliations:** 1Department of Public Health and Clinical Nutrition, University of Eastern Finland, 70210 Kuopio, Finland; ville.m.koistinen@uef.fi (V.M.K.); iman.zarei@uef.fi (I.Z.); kati.hanhineva@uef.fi (K.H.); 2Afekta Technologies Ltd., Yliopistonranta 1L, 70211 Kuopio, Finland; soile.turunen@uef.fi; 3Department of Biochemistry, Food Chemistry and Food Development Unit, University of Turku, 20014 Turku, Finland; 4School of Pharmacy, University of Eastern Finland, 70210 Kuopio, Finland; 5U.S. Department of Agriculture, Agricultural Research Service, Northeast Area, Beltsville Human Nutrition Research Center, Diet Genomics and Immunology Laboratory, Beltsville, MD 20705, USA; gloria.solano-aguilar@usda.gov (G.S.-A.); joe.urban@usda.gov (J.F.U.J.)

**Keywords:** sterol, bile acid, acylcarnitine, LC–MS/MS, annotation

## Abstract

Sterols, bile acids, and acylcarnitines are key players in human metabolism. Precise annotations of these metabolites with mass spectrometry analytics are challenging because of the presence of several isomers and stereoisomers, variability in ionization, and their relatively low concentrations in biological samples. Herein, we present a sensitive and simple qualitative LC–MS/MS (liquid chromatography with tandem mass spectrometry) method by utilizing a set of pure chemical standards to facilitate the identification and distribution of sterols, bile acids, and acylcarnitines in biological samples including human stool and plasma; mouse ileum, cecum, jejunum content, duodenum content, and liver; and pig bile, proximal colon, cecum, heart, stool, and liver. With this method, we detected 24 sterol, 32 bile acid, and 27 acylcarnitine standards in one analysis that were separated within 13 min by reversed-phase chromatography. Further, we observed different sterol, bile acid, and acylcarnitine profiles for the different biological samples across the different species. The simultaneous detection and annotation of sterols, bile acids, and acylcarnitines from reference standards and biological samples with high precision represents a valuable tool for screening these metabolites in routine scientific research.

## 1. Introduction

Sterols, bile acids, and acylcarnitines are important players in the human metabolism. They exert digestive functions, such as hydrolyzing and fermenting complex, undigestible nutrients [1], and also regulate multiple physiological processes in the host [2]. Sterols are a class of lipid molecules which regulate biological processes such as signal transduction, cellular sorting, cytoskeleton reorganization, asymmetric growth, and infectious diseases [3,4]. They have been proposed as key molecules to maintain membranes in a state of fluidity adequate for function [4]. Cholesterol, the major sterol in vertebrates, can be either synthesized by animal cells or obtained from the diet. It helps in maintaining the integrity and fluidity of cell membranes, and serves as a precursor for the synthesis of bile acids, steroid hormones, and vitamin D [5]. Other sterols, such as campesterol, sitosterol, and stanols such as sitostanol, can be derived from food and help in lowering cholesterol levels when the concentrations of serum total and LDL-cholesterol concentrations are high [6,7]. 

Bile acids are amphipathic molecules synthesized in the liver from cholesterol, stored in the gall bladder, and excreted in the intestines in response to the intake of fatty food [8]. The liver synthesizes the primary bile acids such as cholic acid, and bacterial action in the colon synthesizes secondary bile acids such as deoxycholic acid. Bile acids, in general, play a major role in aiding digestion and the absorption of lipids in the small intestine. They also help in the excretion and recirculation of drugs, vitamins, and endogenous and exogenous toxins [8,9]. Moreover, they are endocrine-signaling molecules that regulate metabolic homeostasis through the activation of different receptors, such as the nuclear farnesoid X receptor (FXR) [9,10].

Carnitines, on the other hand, are low-molecular-weight endogenous compounds in most mammalian tissues that play an important role in fatty acid metabolism and cellular energy production [11,12]. Carnitine binds fatty acids, generating various acylcarnitines of different chain lengths to transport activated long-chain fatty acids into the mitochondria for β-oxidation as a major source of energy for cellular activities [13]. Carnitine and acylcarnitines are found in humans, animals, and plants in varying concentrations. 

Over the past decade, many studies have reported altered levels of sterols, bile acids, and acylcarnitines in patients with liver and intestinal diseases [14,15,16]. As an example, elevated levels of serum and fecal bile acids have been observed in patients with Non-Alcoholic Fatty Liver Disease (NAFLD), due to increased bile acid synthesis, increased hepatic bile acids, and upregulation of bile acid transporters [14]. Furthermore, Zhang and Ruitang (2018) suggested a pivotal role of bile acids and the microbial bile acid metabolism as mediators of gut–liver crosstalk subsequently affecting NAFLD initiation and progression [14]. Moreover, decreased levels of plant sterols observed in human serum samples in cross-sectional studies have been associated with NAFLD, while plant sterols purportedly prevent the progression of NAFLD [17,18]. Further, in patients with inflammatory bowel disease including Crohn’s disease and ulcerative colitis, alterations in bile acid metabolism and increased acylcarnitine levels have been observed [16,19]. Hence, the detection of these metabolites may be useful in the evaluation of liver functions and in the diagnosis and treatment of liver and intestinal diseases. These may also be a source for novel biomarker development enabling early disease diagnostics prior to clinical symptoms. 

Gas chromatography–mass spectrometry (GC–MS) and liquid chromatography–mass spectrometry (LC–MS) have been commonly used for the quantitative and qualitative analyses of sterols, bile acids, and acylcarnitines. However, analyzing samples by GC–MS is tedious, as it requires complicated sample preparation procedures encompassing the extraction, purification, and hydrolysis of conjugates [20]. Further, compounds that are non-volatile, polar, or thermally labile need to undergo derivatization to increase their volatility and thermal stability for GC–MS analysis [20]. Hence, LC–MS, with straightforward sample preparation, has become the preferred method for the routine analysis of sterols, bile acids, and acylcarnitines in biological samples. However, the detection of these metabolites is challenging due to the occurrence of many isomers and stereoisomers; and their relatively low concentrations in biological samples [21,22,23]. In addition, the bile acids easily undergo in-source water losses during MS [24]. Hence, the aim of this study was to present and validate a sensitive and simple liquid chromatography–tandem mass spectrometry (LC–MS/MS) method to detect the presence and distribution of sterols, bile acids, and acylcarnitines in biological samples. The approach was based on concurrently analyzing the “Bile acid, Carnitine, and Sterol Library of Standards” (“BACSMLS”, IROA technologies) along with the biological samples in the same analytical run. This semi-targeted method provides a detailed retention time and MS/MS information for the identification library to substantiate the presence of similar compounds in metabolic profiling studies.

## 2. Results and Discussions

### 2.1. Detection and Distribution of Sterol, Bile Acid, and Acylcarnitine Standards

In the present study, we present a simple yet sensitive LC–MS/MS method to detect sterols, bile acids, and acylcarnitines in biological samples. The presented method is simple in terms of sample preparation and LC–MS analysis, while sensitive in precisely detecting multiple compounds exhibiting similar LC–MS characteristics with the employed reversed-phase liquid chromatography–tandem mass spectrometry semi-targeted metabolomics method. Here, in the same LC–MS run, we analyzed the “BACSMLS” metabolite library of standards that contained a total of 94 different sterols, bile acids, and acylcarnitines along with biological samples from humans, mice, and pigs. The sample preparation and LC–MS analysis in this study were based on Klåvus et al. [25], but specifically focused on sterols, bile acids, and acylcarnitines whose precise annotations with mass spectrometry analytics are challenging. The aim was to provide a proof of concept on the applicability of this method for annotating these compound groups in several biological matrices.

In total, 83 standards (24 sterols, 32 bile acids, and 27 acylcarnitines) in the positive ionization mode and 53 standards (3 sterols, 30 bile acids, and 20 acylcarnitines) in the negative ionization mode were detected using this methodology. Appendix A shows the “BACSMLS” standards identified with their LC–MS identification characteristics. Further, the distribution of the different classes of standards, well separated based on retention time in the chromatographic gradient of the applied method, is shown in Figure 1. All the detected sterols, bile acids, and acylcarnitines were separated within 13 min, while the total length of the chromatographic analysis time was 16.5 min. 

A total of 24 sterols out of the 31 different sterol standards in the positive ionization mode were detected using our method. These sterols were found at the end of the chromatogram between 10 and 13 min. However, two more polar C21 sterols (cortisone and corticosterone) eluted earlier than the others (Figure 1). Six sterols from the “BACSMLS” metabolite library that were undetected by our method belonged to the class of cholesteryl esters which are very lipophilic. Hence, they were likely outside our chromatographic range and would have required further adjustment for the non-polar region [26]. In the negative ionization mode, only cortisone, corticosterone, and trace quantities of 7α-hydroxy-4-cholesten-3-one were detected among the sterols.

All 32 bile acids present in the “BACSMLS” library of standards could be detected in the positive ionization mode, including 20 unconjugated, 6 taurine-conjugated, and 6 glycine-conjugated bile acids. They were clustered in the central part of chromatogram between 7 and 10 min (Figure 1). In the negative ionization mode, 30 bile acids out of 32 were detected. Of these, 20 were unconjugated, 5 were taurine-conjugated, and 5 were glycine-conjugated.

Furthermore, 27 out of 31 acylcarnitines were detected. These acylcarnitines had a wide polarity range depending upon the carbon chain length (i.e., different acyl groups) [11,13] and hence were dispersed from the beginning of the chromatogram up to 10 min (Figure 1). Hydrophilic and short-chain acylcarnitines (propionylcarnitine, acetyl-DL-carnitine, succinylcarnitine, hydroxybutyrylcarnitine, and glutarylcarnitine) had poor retention on a reversed-phase column, as represented by an early eluting poor peak shape. In the negative mode, 20 acylcarnitines were detected but 15 of these were detected in trace quantities. 

#### 2.1.1. Sterol Characteristics in LC–MS/MS

The extracted ion chromatograms (EICs) of the base peak spectra of the sterols detected in the positive ionization mode are shown in Figure 2. In the positive mode, the most abundant signal in the mass spectrum for most of the sterols was the [M+H–H_2_O]^+^ ion. For sterols with two OH groups, the [M+H–2H_2_O]^+^ ion was also visible. The mass spectra of cortisone, corticosterone, diosgenin, 7α-hydroxy-4-cholesten-3-one, and 7-ketocholesterol showed abundant [M+H]^+^ ions (Appendix A).

In general, increasing the degree of unsaturation in the B-ring or in the side chain resulted in shorter retention times (Figure 2). Similar observations for non-isomers were reported by Münger et al. [27]. The chromatographic separation resolved oxysterol isomers, including 24- and 27-hydroxycholesterol, as well as 7α- and 7β-hydroxycholesterol. A high-intensity fragment of 24-hydroxycholesterol was displayed at *m*/*z* 367.3355, which was present in very low intensity in 27-hydroxycholesterol (Figure 3). 

#### 2.1.2. Bile Acid Characteristics in LC–MS/MS

Figure 4A–C show the EICs of the base peak spectra of the bile acids in the positive ionization mode. The most abundant ion in the mass spectrum for most of the bile acids was [M+H–H_2_O]^+^ ion. Bile acids with identical *m*/*z* were distinguished from one another solely based on differences in their retention times, which in turn were influenced by several factors. Firstly, the bile acid skeleton and the side chain structures influenced the hydrophilicity and hence their elution order [28]. The order of elution was as follows: tri-hydroxy bile acids (e.g., cholic acid) < di-hydroxy bile acids (e.g., deoxycholic acid) < mono-hydroxy bile acids (e.g., lithocholic acid) (Figure 4D).

Secondly, the position and stereochemistry of the hydroxyl groups also influenced the retention times of the bile acids [28]. As an example, ursodeoxycholic acid, which is a di-hydroxy bile acid, eluted before cholic acid, which is a tri-hydroxy bile acid. This is because the b-hydroxyl group of ursodeoxycholic acid is located above the hydrophobic surface, thereby weakening its bonding with the stationary phase and causing earlier elution [29]. In the case of di-hydroxy bile acids, the order of elution was as follows: ursodeoxycholic acid < chenodeoxycholic acid < deoxycholic acid (Figure 4E). Similar results were also obtained from a study by Huang et al. [28], who attributed this to the orientation of the hydroxyl substitutions and their ability to form intra-molecular H-bonding. 

In general, for a specific bile acid skeleton, the elution order in reversed-phase chromatography was as follows: taurine-conjugated, glycine-conjugated, and unconjugated forms [29]. As an example, taurodeoxycholic acid eluted first, followed by glycodeoxycholic acid, and then deoxycholic acid (Figure 4F).

Further, the identification of bile acids in mass spectrometry is often challenged by the relatively easy in-source fragmentation of water molecules from many of the structures. The alignment spot viewer in MS-DIAL proved to be helpful in indicating peaks corresponding to in-source fragments which were then further verified by manual inspection of the raw data in the EICs and MS/MS spectra. Figure 5 shows the EICs of six different bile acids. They all have the same neutral mass of 408.287 but occur with several different base peak ions, either as a dimer, or after loss of one or more water molecules.

#### 2.1.3. Acylcarnitines Characteristics in LC–MS/MS

The EICs of the base peak spectra of the detected acylcarnitines are shown in Figure 6. The most abundant ion in the mass spectrum for all the acylcarnitines was the [M + H]^+^ ion. Further, the fragmentation patterns of all the acylcarnitines had prominent fragment ions at *m*/*z* 85.028 and 60.081 in common, representing the polar head group with trimethylated quaternary ammonium. 

The retention time increased linearly as the number of carbon atoms in the molecule increased and decreased with the increase in the number of double bonds. Moreover, using our method, isomeric acylcarnitines were distinguished solely based on the retention times. As an example, three isomeric acylcarnitines (with monoisotopic mass 245.162 Da) eluted in the following order: 2-methylbutyroylcarnitine, followed by iso-valeryl carnitine and finally valeryl carnitine (Figure 6 and Figure 7), in accordance with the rule that branched carbon chains elute first [30]. However, these three acylcarnitines had the same MS/MS fragmentation pattern and could only be distinguished based on the elution order (Figure 7).

### 2.2. Distribution of Sterol, Bile Acid, and Acylcarnitine Profiles in Biological Samples

The LC–MS/MS method was applied to a series of samples including human stool and plasma; mouse ileum, cecum, jejunum content, duodenum content, and liver; and pig bile, proximal colon, cecum, heart, stool, and liver to study the distribution of sterols, bile acids, and acylcarnitines in biological samples. The “BACSMLS” standards present in the biological samples were precisely annotated, including the closely eluting structural isomers, and their distributions verified in various sample matrices across three different species (Figure 8A,B, and Appendix A). No confirmed cases of matrix effect were observed between the different sample matrices on retention times beyond 0.1 min (Figure 8B, Appendix A).

After the annotation of the “BACSMLS” standards in the biological samples, multivariate statistics was applied to explore their distribution sample-wise. In the principal component analysis (PCA) representations of the samples (Figure 9), most of the sample types were well separated based on their sterol, bile acid, and acylcarnitine profiles. The score plots of the PCA revealed that 24.5% and 19.9% of the variation were explained by PC1 and PC2, respectively. This established PCA model also highlighted clustering between the same sample types from different species, solely based on the sterol/bile acid/acylcarnitine profiles. As an example, the stool samples from humans and pigs clustered together with the intestinal contents from mouse. Furthermore, the intestinal tissues such as cecum, ileum, and proximal colon from mice and pigs clustered together. In addition, the heart tissue from pig clustered with human plasma. The pig bile, which had a different sterol, bile acid, and acylcarnitine profile, clustered separately from the rest of the biological samples.

In addition to PCA, the metabolite distribution across the different sample types was examined using Pearson correlation, and a similar observation was seen on the correlation plot (Figure 10). Human plasma and pig heart were well correlated, while the pig bile and liver metabolite content showed strong correlation. The jejunum content of mouse was strongly correlated with the duodenum content of mouse. The intestinal contents of mouse were correlated to a lesser extent to human stool and mouse cecum samples. Further, the pig proximal colon correlated strongly with the cecum tissue of pig and pig stool, and to a lesser extent with the mouse cecum tissue and human stool samples. Interestingly, correlations between mouse liver and tissue samples from ileum, cecum and proximal colon of mouse and pigs were observed but the liver samples from mouse did not correlate well with the liver samples from pig.

The number of the sterol, bile acid, and acylcarnitine metabolites identified in the biological samples from humans, mice, and pigs varied, as illustrated in the heatmap in Figure 11 and Table 1. 7α-hydroxy-4-cholesten-3-one, oleoylcarnitine, octadecanoylcarnitine, and 7-ketocholesterol (5-cholesten-3β-ol-7-one) were present in all samples. Cholesterol, myristoyl-L-carnitine, and hexadecanoylcarnitine were detected in all samples except pig stool. Octenoyl-L-carnitine was detected only in pig stool.

#### 2.2.1. Sterol/Bile Acid/Acylcarnitine Profiles in Liver Tissues

A comparison made of the sterol/bile acid/acylcarnitine profiles in the liver tissues of mouse and pig revealed the predominance of taurine conjugated bile acids in mouse liver in contrast to pig liver. This could be attributed to the enzyme in mice for bile acid conjugation which is specific to taurine, rather than glycine conjugation [22,31]. Glycohyodeoxcholic acid and glycochenodeoxycholic acid, two major bile acids in the pig liver, were absent in mice, whereas the taurine conjugate of muricholic acid (tauromuricholic acid) was the most abundant bile acid in the mouse liver and was not present in the pigs’. Further, marked differences in the relative abundances of certain acylcarnitines, such as glutarylcarnitine, octanoyl-L-carnitine, 3-hydroxyhexadecanoylcarnitine, and acetyl-L-carnitine, were observed. Moreover, as previously described by Tsai et al. [32], cholesterol was the major sterol component in the liver samples of both pigs and mice.

#### 2.2.2. Sterol/Bile Acid/Acylcarnitine Profiles in Plasma and Heart Samples

The sterol, bile acid, and acylcarnitine profiles of human plasma were similar to that of pig heart. In particular, cholesterol gave a very clear signal in both sample types. Consistent with previous studies, 7α-hydroxycholesterol, the most abundant oxysterol in plasma, also gave a clear signal [33,34]. It is an intermediate in the metabolism of cholesterol and plays an important role in cholesterol homeostasis [35]. However, two other abundant oxysterols (27- and 24-hydroxycholesterol) were found in trace quantities. Moreover, in both human plasma and pig heart, several acylcarnitines (10 in plasma, 12 in heart) were detected, presumably because they readily cross mitochondrial and cell membranes (unlike their corresponding acyl-CoA analogues), and are detectable in plasma, thereby giving a reflection of the intramitochondrial acyl-CoA status at the time of analysis [36].

#### 2.2.3. Sterol/Bile Acid/Acylcarnitine Profiles in Intestinal Tissues and Contents 

The sterol/bile acid/acylcarnitine composition was further investigated in the intestinal tissues of mice and pigs, intestinal contents of mice, and stools of humans and pigs. As bile acids are transformed by gut microbiota, many secondary bile acids, such as deoxycholic acid and hyodeoxycholic acid were observed as clear signals. Further, due to 7α-dehydroxylation [37] and epimerization of chenodeoxycholic acid [38], ursodeoxycholic acid was also found. However, in mice, ursodeoxycholic acid is considered a host-derived primary bile acid [39]. Lithocholic acid, another secondary bile acid, gave clear signals in human and pig samples. This bile acid, however, was not found in mouse samples. This is consistent with the findings of Eyssen et al. who reported trace quantities of lithocholic acid in mouse gut by GC–MS [40]. 

The sterol, bile acid, and acylcarnitine profiles of mouse jejunum and duodenum were similar to each other. However, their relative abundance varied. Likewise, the sterol/bile acid/acylcarnitine profile of mouse cecum and ileum tissues were similar; however, tauro-conjugated bile acids gave better signals in ileum than cecum. This could be due to the fact that most amidated bile acids are actively absorbed in the ileum [41]. Furthermore, certain acylcarnitines such as 2-methylbutyroylcarnitine, isovaleryl-L-carnitine, and valeryl-L-carnitine gave clearer signals in cecum tissues than ileum tissues. Although the relative abundances varied, the bile acid/sterol/carnitine profile of pig proximal colon and cecum tissues were similar to each other. The major difference between the bile acid/sterol/carnitine profile of the intestinal tissue samples of mice and pigs was that the mouse samples had predominantly tauro-conjugated bile acids and muricholic acid. Similar observations of increased tauro-conjugated bile acid and muricholic acid concentrations in mice was reported by Li et al. [42] and Eyssen et al. [40], respectively. Many sterols, bile acids, and acylcarnitines could be detected in the fecal samples of humans and pigs. Although chenodeoxycholic acid and cholic acid are primary bile acids in both humans and pigs [43], only chenodeoxycholic acid gave a clear signal in both samples, whereas cholic acid was only observed in human stool samples. It had been reported earlier that cholic acid decreases in the late gestation period and the proportion of another trihydroxy bile acid, hyocholic acid, is found in levels equal to that of cholic acid in humans [44]. However, in our study, hyodeoxycholic acid gave a signal in pig stool samples, presumably due to the contribution of gut microbiota [44]. Further, other metabolites, such as 7-ketodeoxycholic acid, coprocholic acid (3α,7α,12α-trihydroxycoprostanic acid), hexadecanoylcarnitine, and myristoyl-L-carnitine, also gave clear signals in human stool samples but were not present in pig feces.

#### 2.2.4. Sterol/Bile Acid/Acylcarnitine Profiles in Bile Samples

The highest number of sterol, bile acid, and acylcarnitine profiles were found in the pig bile samples, and they differed from the other biological samples analyzed. Bile is mainly constituted by bile acids, cholesterol, phospholipids, and proteins [45]. Clearly, the pig bile analyzed in our study showed clear signals of many glyco- and tauro-conjugated bile acids, and cholesterol. In particular, glycodeoxycholic acid and glycocholic acid gave clear signals in pig bile compared to other sample types. Moreover, pig bile showed two peaks for *m*/*z* 931.625—a major peak at 8.1 min and a minor peak at 8.5 min. The minor peak in the sample aligned with the glycocholic acid peak in the standard. However, the major peak could not be annotated based on our list of “BACSMLS” standards analyzed. This could have corresponded to another isomer such as glycohyocholic acid, which was not analyzed in our study (data not shown). Furthermore, no spectral match in the METLIN online database was found [46].

### 2.3. Strengths and Limitations of the Study

In this study, we presented a LC–MS/MS method for simultaneous detection of sterols, bile acids, and acylcarnitines from various biological samples to precisely annotate the detected “BACSMLS” standards across three metabolite classes, including isomeric compounds. We utilized structure-based elution order to correctly annotate the isomeric peaks. We could further observe different sterol, bile acid, and acylcarnitine profiles for the various biological samples. This method will enable the automated annotation of many compounds with challenging ionization and in-source fragmentation such as bile acids.

However, this study has some limitations. Firstly, the methods employed do not accurately quantify the compounds (in terms of concentration) but rather provide relative amounts of sterol/bile acid/acylcarnitine (e.g., their profiles) in biological samples. Secondly, only 83 of 94 “BACSMLS” standards were detected due to possible limitations in the analytical coverage of our method. Thirdly, some of the acylcarnitines that were analyzed (propionyl carnitine, acetyl-DL-carnitine, succinylcarnitine, hydroxybutyrylcarnitine, and glutarylcarnitine) were very hydrophilic and had poor retention on the RP column. Hence, in future analyses, the HILIC mode could be employed in parallel to extend the analytical range to these compounds. 

Further, the biological sample size was relatively small and, in most cases, had only two samples from representative sample types from each species. Biological or technical replicates were not included because the aim was simply to determine if our method could detect the presence of “BACSMLS” standards in our samples. Moreover, the sex of the individuals can largely influence the baseline metabolome [47]. However, the assessment of the gender-specific effects was not in the focus of our current analysis. In this study, the different standards, including isomers in our samples, were detected and differences between the species and sample types were distinguished by the sterol/bile acid/acylcarnitine profiles. There was no confirmed case of matrix effect observed between the different sample types beyond 0.1 min. Further, although the 94 unique “BACSMLS” standards used in this study included small molecule organic acid metabolites covering key metabolic pathways, there are many other sterols, bile acids, and acylcarnitines in biological samples that were not considered. Nevertheless, the MS/MS fragmentation profiles obtained from the pure compounds used here can be utilized to annotate similar compounds in the same compound classes in future studies. 

## 3. Materials and Methods

### 3.1. Reagents

A set of 96 “Bile acid, Carnitine, and Sterol library of standards” (“BACSMLS”; 31 sterols, 33 bile acids, and 32 acylcarnitines) as shown in Appendix A, were obtained from IROA technologies (Sea Girt, NJ, USA). Each well of the “BACSMLS” standard plate contained 5 µg (at least 95% purity) of each compound. Two of the standards—propionyl-L-carnitine/propanoylcarnitine and tauromuricholic acid/tauro-β-muricholic acid—were found to be duplicates (identical KEGG/HMDB ID and CAS ID provided by the supplier). 

All solvents and chemicals used were of either high performance liquid chromatography (HPLC) grade or of known analytical purity. Methanol and isopropanol were obtained from CHROMASOLV LC–MS Ultra, Riedel-de Haën, Honeywell, Seelze, Germany. Ethanol was obtained from Altia Industrial Services, Rajamäki, Finland. Acetonitrile was obtained from HiPerSolv CHROMANORM, VWR Chemicals, Fontenay-sous-Bois, France. Formic acid was obtained from Optima LC/MS, Fisher Chemical, Geel, Belgium, Cat.No. A117-50. Ultra-pure water (Class 1, ELGA PURELAB Ultra Analytical, Lane End, UK) was used in this study.

### 3.2. Standard Preparation

Most of the standards were solubilized individually using 250 µL HPLC grade methanol. Seven standards had poor solubility in methanol and, therefore, 5α-cholestane, ergosterol, and cholic acid were solubilized in 250 µL ethanol solution; and cholesteryl palmitate, cholesteryl oleate, cholesteryl stearate, and cholesteryl behenate were solubilized in 250 µL isopropanol solution [31]. The standards were kept at 4 °C until analysis. Assuming that the standards were solubilized completely, the theoretical concentration of the spiked standards would be 20 µg/mL.

### 3.3. Sample Collection

A total of 13 sample types were analyzed in this study, namely, plasma and stool from humans; bile, heart, proximal colon, cecum, liver, and stool from pigs; and duodenum content, jejunum content, liver, ileum, and cecum from mice. 

The human plasma samples were obtained from four healthy male subjects (mean ± standard deviation: 57.3 ± 3.5 years) with unchanged dietary habits [48]. Plasma samples were separated from the venous blood samples and stored at −80 °C. The human stool samples were obtained from two male subjects (mean ± standard deviation: 64.5 ± 12.02 years) with unchanged dietary habits (Babu, Csader et al. submitted manuscript). Stool samples were collected in a plastic container with lid by the subject himself while wearing gloves. The sealed container was placed in an icebox filled with ice bags and brought to the research unit the next day. At the research unit, stool samples were directly homogenized, aliquoted, and frozen at −80 °C without any detergents.

The cecum and ileum tissue samples were obtained from two healthy 17-week-old C57BL/6J male mice fed a low-fat diet. The mice were fasted for 7.5–8.5 h and sacrificed by decapitation after being made unconscious by CO_2_ gas. The tissue samples were dissected immediately after blood collection, rinsed with physiological saline, wrapped in aluminum foil, snap frozen in liquid nitrogen, and stored at −70 °C.

The mouse plasma and luminal contents were obtained from two healthy 14-week-old male JaxC57BL/6J (Jackson Laboratory, Bar Harbor, ME, USA) mice fed a standard chow diet. The mice were first placed under inhalation anesthesia with isoflurane via an induction chamber and maintained using mask anesthesia followed by heart blood sample collection via cardiac puncture and terminated using cervical dislocation. Plasma samples were immediately separated from the whole blood using plasma tubes with separating gel and lithium heparin according to the manufacturer’s instructions, aliquoted, and frozen in liquid nitrogen before being stored at −80 °C. The liver was collected and weighed immediately after termination, and snap frozen immediately in liquid nitrogen and stored at −80 °C. Sterile L-shaped plate spreader was used to remove the luminal contents onto a sterile aluminum foil. The contents were then transferred into an Eppendorf tube, which was snap frozen directly in liquid nitrogen before being stored at −80 °C.

The pig samples were obtained from two 16-week-old healthy female conventional pigs (Landrace X Yorkshire) fed a standard pig grower diet. They were bred at the US Department of Agriculture—Beltsville Swine Unit. The samples were collected immediately after the pigs were euthanized with a pentobarbital-based solution (Euthasol Virbac Animal Health). After the removal of the complete organs, representative samples from proximal colon, cecum, liver, and heart tissues were cut and placed into sample collection tubes and snap frozen immediately in liquid nitrogen before being stored at −70 °C. Bile samples were collected with syringe and needle and placed in collection tubes that were snap frozen in liquid nitrogen before being stored at −70 °C. Stool samples were collected from lumen of the proximal and distal colon and placed into collection tubes before snap freezing in liquid nitrogen and storage at −70 °C.

### 3.4. Sample Preparation

The stool, heart, proximal colon, cecum, liver, duodenum content, jejunum content, and ileum were homogenized by adding 80% *v*/*v* aqueous HPLC grade methanol in a ratio of 500 µL per 100 mg of sample for the metabolite extraction and protein precipitation using Bead Ruptor 24 Elite homogenizer at the speed 6 m/s at 0 ± 2 °C for 30 s. The samples were subsequently vortexed and centrifuged for 10 min at 4 °C and 20,000× *g*. The supernatant was collected and filtered (0.2 µm PTFE membrane) into HPLC vials and kept at 4 °C until analysis. 

Plasma samples were prepared by adding 400 µL of acetonitrile (kept at 4 °C) to 100 µL of plasma, and bile samples were prepared by adding 500 µL of cold acetonitrile per 100 µL of sample. The precipitated samples were filtered (Captiva ND filter plate 0.2 µm) by centrifuging for 5 min at 700× *g* at 4 °C. The filtered samples were transferred into HPLC vials and kept at 4 °C until analysis. The standards and biological samples were analyzed in the same analytical run. 

### 3.5. LC–MS Analysis

The liquid chromatography–mass spectrometry (LC–MS) analysis was performed according to Klåvus et al. [25]. Briefly, liquid chromatography was performed on a 1290 Infinity Binary UPLC (Agilent Technologies). Reversed-phase separation was employed using a Zorbax Eclipse XDB-C18 column (dimensions 2.1 × 100 mm, particle size 1.8 µm) as the stationary phase. The mobile phase consisted of ultra-pure water (solution A) and methanol (solution B), both containing 0.1% *v*/*v* formic acid. The flowrate of the eluent was 0.6 mL/min. The elution gradient profile was as follows (t [min], %B): (0, 2), (10, 100), (14.5, 100), (14.51, 2), (16.5, 2). A measure of 1 µL of each standard or sample was injected for analysis. 

Mass spectrometry was performed on an Agilent 6540 Q-TOF with a Jet Stream ESI ion source. The fragmentor voltage used was 100 eV and scan range 20–1600 *m*/*z*. From every precursor scan cycle (400 milliseconds), four most abundant precursor ions were automatically selected for MS/MS fragmentation, excluded after two acquired MS/MS spectra, and released again from the exclusion list after 0.25 min. The collision energies used for the MS/MS analysis were (±)10, 20, and 40 eV, for compatibility with online databases such as METLIN and HMDB database [46,49]. Both negative and positive ionization modes were used in the study. 

### 3.6. Data Analysis

After collection of the raw instrumental data, MS-DIAL (Version 4.18) [29] was employed for automated peak picking and alignment from the raw data. The minimum peak height was set to 2000, MS/MS tolerance was 0.05, and RT tolerance was set to 0.1 min. Default settings were used for the other parameters. After peak picking, the alignment result as peak areas were exported into Microsoft Excel. The exported excel file contained a data matrix comprising the LC–MS characteristics along with the peak areas of the different standard compounds and the biological samples. The raw spectral data from the pure compound standards and biological samples were manually examined and the unique peaks corresponding to each of the individual standard compounds were located, followed by annotation in the data matrix. Thereafter, the spectral base peak corresponding to each standard was annotated as the representative signal for that standard. Moreover, the typical adduct and neutral loss peaks were annotated including [M+H]^+^, [M+H–H_2_O]^+^, [M+H–2H_2_O]^+^, [2M+H]^+^, [M+NH_4_]^+^, and [M+Na]^+^ for the positive ionization mode and [M–H]^−^, [M–H–H_2_O]^−^, [2M–H]^−^, and [M+FA–H]^−^ for the negative ionization mode. Further, MS/MS data from METLIN database [46] and HMDB [49] were used as additional examination for the fragmentation of some of the standards. The vendor software—Agilent MassHunter Qualitative Analysis B.07.00—was used for the exploration of raw data extracted ion chromatograms (EICs) and MS/MS fragmentation spectra of the pure standards and biological samples. The limit of detection of the metabolites was set by following two criteria: signal to noise ratio >5 and raw peak area >10,000. All those signals with a peak area <10,000 were denoted as trace. 

After annotating the respective signals for each of the pure standard compounds across all the biological samples, the raw base peak area values obtained from the MS-DIAL analysis were used to determine the relative levels of the compounds in the samples. These values were then averaged across each sample type. Multivariate statistical analysis was performed in RStudio Software (Version 3.6.2). The freely downloaded version of Multiple Array Viewer (MeV; version 4.9.0) (http://www.mev.tm4.org/) (accessed on 28 May 2021) was used for further visualization, data normalization, and hierarchical clustering analyses. All visualizations presented in this paper used data from the positive ionization mode.

## 4. Conclusions

A simple and effective LC–MS/MS method to detect the distribution of 83 sterols, bile acids, and acylcarnitines in biological samples was presented. The simultaneous detection of sterols, bile acids, and acylcarnitines from reference standard samples and biological samples allows us to annotate them with high precision in the biological samples and represents a valuable tool for screening sterol, bile acid, and acylcarnitine profiles in routine scientific research.

## Figures and Tables

**Figure 1 metabolites-12-00049-f001:**
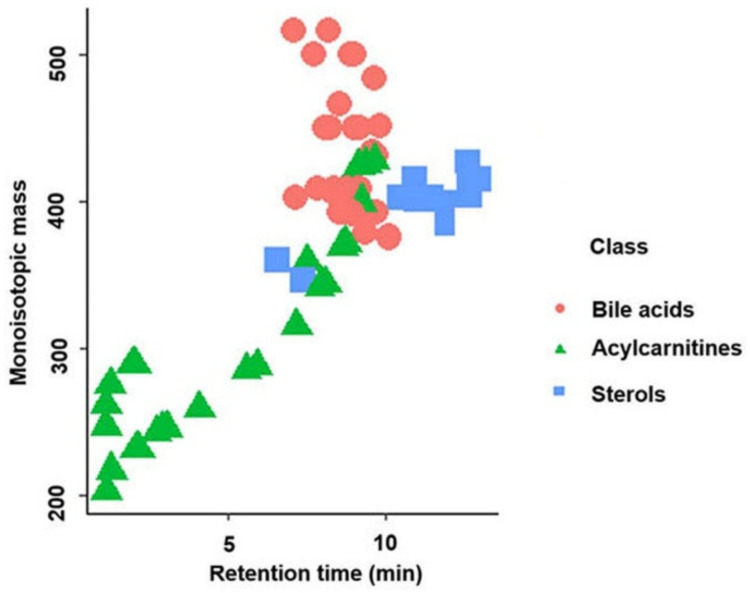
Separation of the different classes of standards (sterols, bile acids, and acylcarnitines) based on retention time in reversed-phase liquid chromatography in the positive ionization mode.

**Figure 2 metabolites-12-00049-f002:**
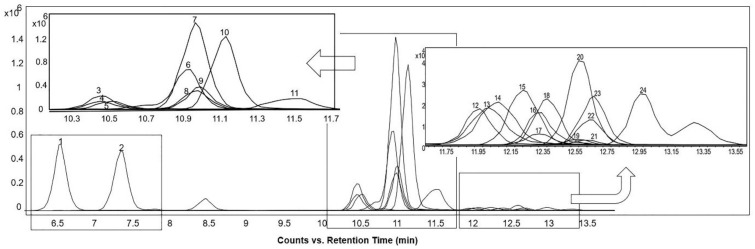
Extracted ion chromatograms of the base peak spectra of detected sterols acquired from reference “BACSMLS” standards in the positive ionization mode. Peak identities are as follows: (1) cortisone, (2) corticosterone, (3) 25-hydroxycholesterol, (4) 24-hydroxycholesterol, (5) 27-hydroxycholesterol, (6) diosgenin, (7) 7α-hydroxy-4-cholesten-3-one, (8) 7α-hydroxycholesterol, (9) 7β-hydroxycholesterol, (10) 7-ketocholesterol (5-cholesten-3β-ol-7-one), (11) cholesterol 5α,6α-epoxide, (12) desmosterol, (13) ergosterol, (14) 7-dehydrocholesterol, (15) brassicasterol, (16) 5β-cholestan-3α-ol (epicoprostanol), (17) lathosterol, (18) cholesterol, (19) coprostanol (coprostan-3-ol), (20) lanosterol, (21) dihydrocholesterol (5α-cholestan-3β-ol), (22) stigmasterol, (23) campesterol, (24) β-sitosterol.

**Figure 3 metabolites-12-00049-f003:**
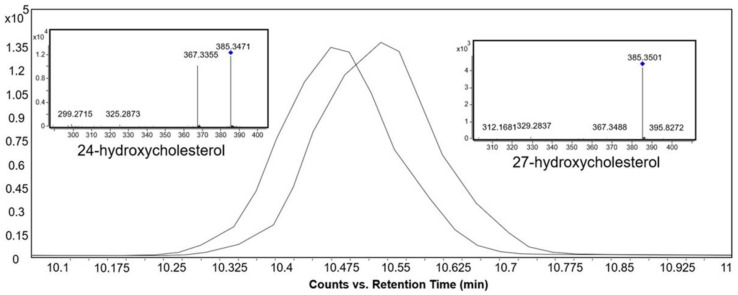
Extracted ion chromatograms and MS/MS spectra of the base peak spectra of 24-hydroxycholesterol and 27-hydroxycholesterol standards at 10 eV collision energy in the positive ionization mode.

**Figure 4 metabolites-12-00049-f004:**
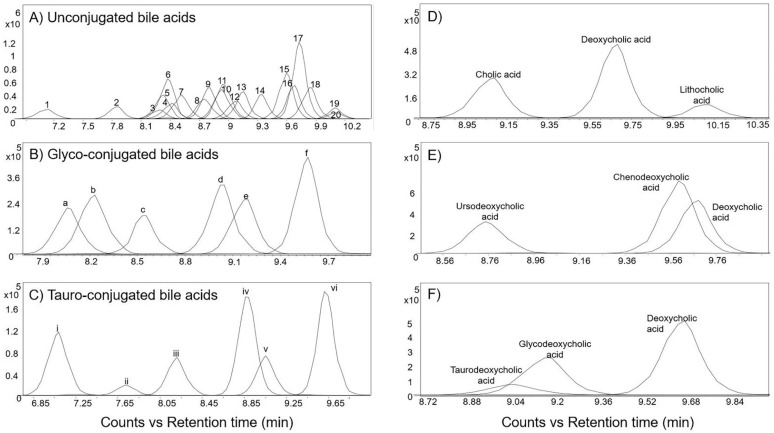
Extracted ion chromatogram of the base peak spectra of bile acids acquired from reference “BACSMLS” standards in the positive ionization mode. (**A**) EICs of unconjugated bile acids. Peak identities are as follows: (1) dehydrocholic acid, (2) ursocholic acid, (3) 7-ketodeoxycholic acid (7-keto-3α,12-α-dihydroxycholanic acid), (4) 12-ketochenodeoxycholic acid ((3alpha,5beta,7alpha)-3,7-dihydroxy-12-oxocholan-24-oic acid), (5) alpha-muricholic acid, (6) beta-muricholic acid, (7) murideoxycholic acid, (8) ursodeoxycholic acid, (9) gamma-muricholic acid (hyocholic acid), (10) 7-ketochenodeoxycholic acid (nutriacholic acid), (11) hyodeoxycholic acid, (12) cholic acid, (13) A=allocholic acid, (14) nordeoxycholic acid, (15) chenodeoxycholic acid, (16) deoxycholic acid, (17) 7α-hydroxy-3-oxo-4-cholestenoic acid (7-HOCA), (18) coprocholic acid (3α,7α,12α-trihydroxycoprostanic acid), (19) dehydrolithocholic acid (3-oxo-5β-cholanoic acid), (20) lithocholic acid. (**B**) EICs of glycine-conjugated bile acids acquired from reference “BACSMLS” standards—(a) Glycoursodeoxycholic acid, (b) glycohyodeoxcholic acid, (c) glycocholic acid, (d) glycochenodeoxycholic acid, (e) glycodeoxycholic acid, (f) glycolithocholic acid. (**C**) EICs of taurine-conjugated bile acids—(i) tauromuricholic acid (ii) tauroursodeoxycholic acid, (iii) taurocholic acid, (iv) taurochenodesoxycholic acid, (v) taurodeoxycholic acid, (vi) taurolithocholic acid. (**D**) Order of elution of bile acids influenced by bile acid nucleus and the side chain structures. (**E**) Order of elution of bile acids influenced by the position and stereochemistry of hydroxyl groups. (**F**) Order of elution based on the bile acid conjugation.

**Figure 5 metabolites-12-00049-f005:**
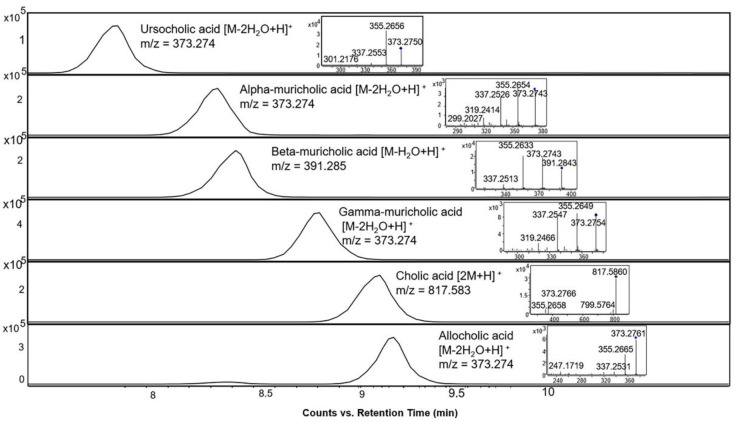
Extracted ion chromatograms and MS/MS spectra of bile acid standards with different representative base peak ions in the positive ionization mode at 10 eV collision energy, all of which have a neutral mass of 408.287.

**Figure 6 metabolites-12-00049-f006:**
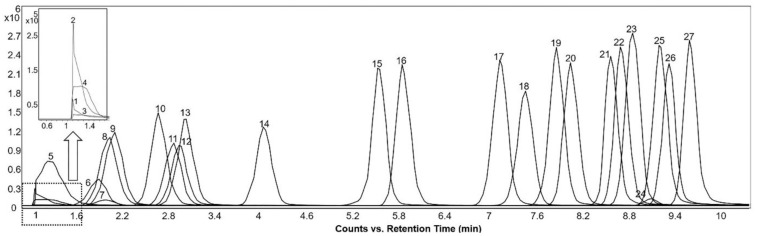
Extracted ion chromatograms of the molecular ions of detected acylcarnitines acquired from reference “BACSMLS” standard samples in the positive ionization mode. Peak identities are as follows: (1) succinylcarnitine, (2) hydroxybutyrylcarnitine (3) acetyl-DL-carnitine, (4) glutarylcarnitine, (5) propionyl-L-carnitine, (6) adipoyl-L-carnitine, (7) methylglutaryl-L-carnitine, (8) isobutyryl-L-carnitine, (9) butanoylcarnitine (Butyryl-L-carnitine), (10) tiglylcarnitine, (11) 2-methylbutyroylcarnitine, (12) isovaleryl-L-carnitine, (13) valeryl-L-carnitine, (14) hexanoylcarnitine, (15) octenoyl-L-carnitine, (16) octanoyl-L-carnitine, (17) decanoyl-L-carnitine, (18) 3-hydroxydodecanoylcarnitine, (19) dodecenoylcarnitine, (20) dodecanoylcarnitine (lauroyl-L-carnitine), (21) tetradecenoyl-L-carnitine, (22) myristoyl-L-carnitine, (23) 3-hydroxyhexadecanoylcarnitine, (24) linoleylcarnitine (cis,cis-9,12-octadecadienoyl-L-carnitine), (25) hexadecanoylcarnitine (palmitoyl-L-carnitine), (26) oleoylcarnitine, (27) octadecanoylcarnitine (stearoyl-L-carnitine).

**Figure 7 metabolites-12-00049-f007:**
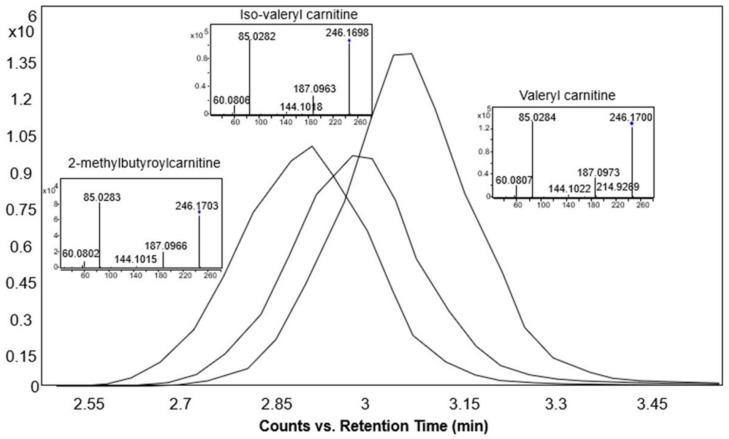
Extracted ion chromatograms and MS/MS spectra of the base peak spectra of 2-methylbutyroylcarnitine, iso-valeryl carnitine, and valeryl carnitine standards at 10 eV collision energy in the positive ionization mode.

**Figure 8 metabolites-12-00049-f008:**
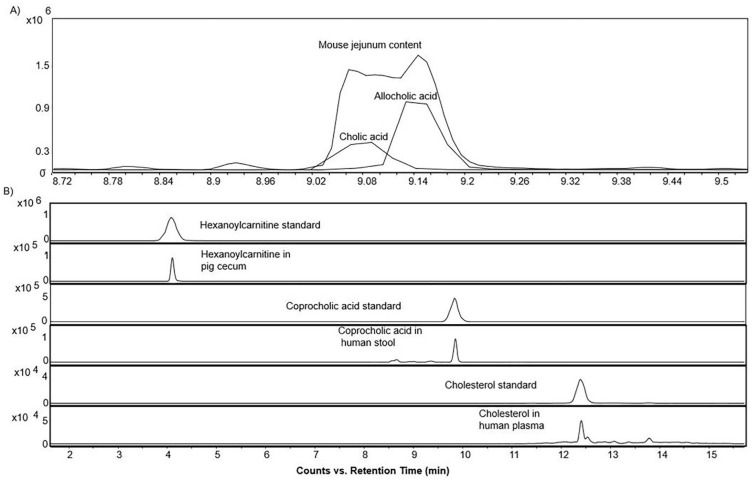
(**A**) Extracted ion chromatograms of the base peak spectra of bile acid isomer (cholic acid and allocholic acid) and their presence in mouse jejunum content in the positive ionization mode (**B**) Extracted ion chromatograms of the base peak spectra of a sterol, bile acid, and acylcarnitine standard, and their occurrence in the biological samples in the positive ionization mode demonstrating the absence of any confirmed case of matrix effect beyond 0.1 min.

**Figure 9 metabolites-12-00049-f009:**
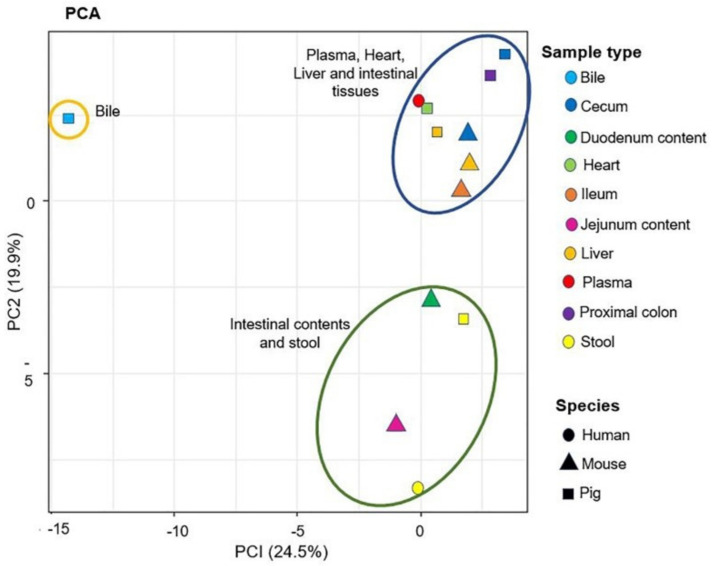
Principal component analysis (PCA) of the biological samples analyzed in the positive ionization mode. This figure contains the first two principal components and their scores PC1 and PC2, which explain 24.5% and 19.9% of the variation within the data, respectively. The shapes denote the different species and the colors denote the sample types. The plot shows three clusters (statistical significance not evaluated): blue ellipse—plasma, heart, liver, and intestinal tissues; green ellipse—intestinal content and stool samples; brown ellipse—bile.

**Figure 10 metabolites-12-00049-f010:**
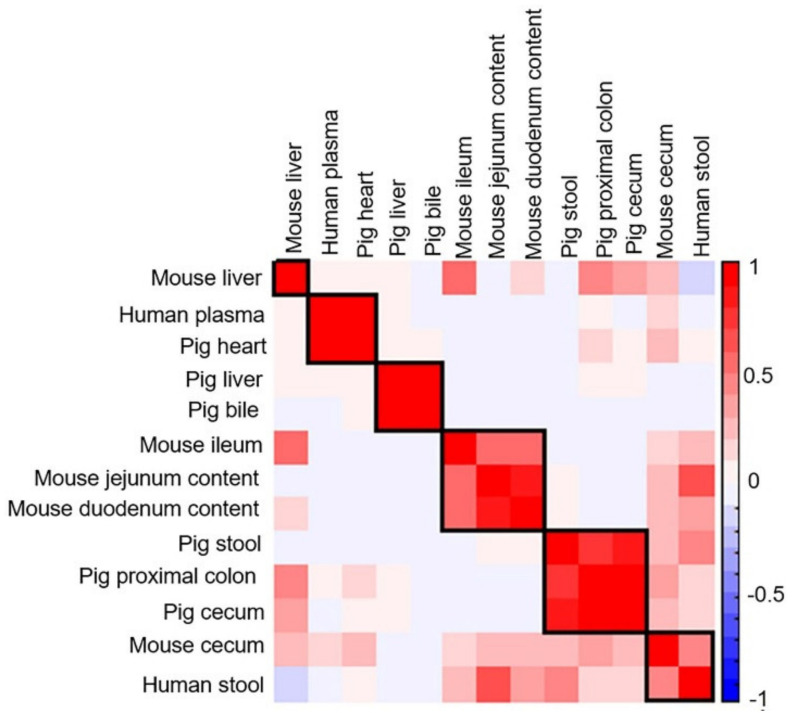
Correlation matrix of analyzed biological samples in the positive ionization mode. Values and shading intensities represent Pearson rank correlation coefficients.

**Figure 11 metabolites-12-00049-f011:**
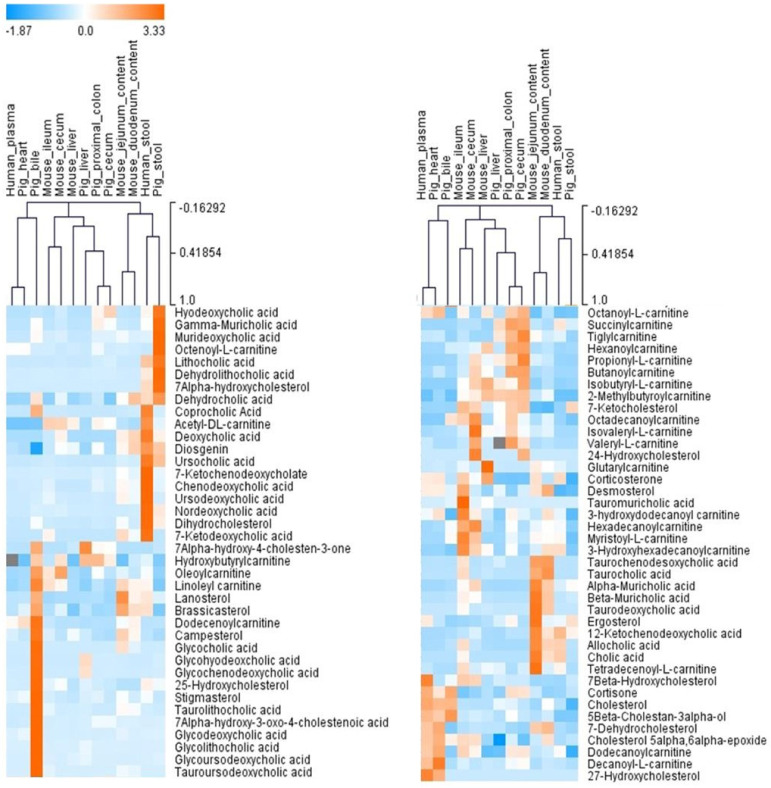
An unsupervised hierarchical clustering analysis based on the qualitative analysis of “BACSMLS”. Each row shows the data for a specific sterol/bile acid/acylcarnitine, and each column shows the “BACSMLS” profiles for the different biological samples. All the identified metabolites were included. Clustering was performed for both metabolites and biological samples (dendrograms not shown for metabolites). The clustering analysis efficiently distinguishes between the different sample types and, more importantly, between the studied species. The raw peak area values from MS-DIAL were used to determine the relative levels of the compounds in the biological samples. For isomers, this value was manually integrated in the vendor software—Agilent MassHunter Qualitative Analysis B.07.00. The peak areas were normalized across all compounds and samples using the mean and the standard deviation (Z-score).

**Table 1 metabolites-12-00049-t001:** Total number of sterol, bile acid, and acylcarnitine metabolites identified in biological samples.

Species	Sample Type	Sterol	Bile Acid	Acylcarnitine	Total
Human	Stool	4	24	9	37
	Plasma	5	4	10	19
Mouse	Duodenum content	5	23	16	44
	Jejunum content	5	24	14	43
	Ileum	5	20	17	42
	Cecum	4	15	15	34
	Liver	3	7	17	27
Pig	Bile	9	20	10	39
	Proximal colon	4	13	19	36
	Cecum	4	12	18	34
	Stool	4	21	3	28
	Liver	4	6	14	24
	Heart	4	3	12	19

## Data Availability

Access to the raw data, R codes, and/or material can be sought via contacting the responsible authors. The data are not publicly available due to privacy.

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
