# Peer review of "Identification and Distribution of Sterols, Bile Acids, and Acylcarnitines by LC–MS/MS in Humans, Mice, and Pigs—A Qualitative Analysis"

_metabolites, 2022, doi:10.3390/metabo12010049_

Round 1

Reviewer 1 Report

We have reviewed the manuscript titled “Identification and Distribution of Sterols, Bile Acids, and Acyl- 2 carnitines by LC-MS/MS in Humans, Mice, and Pigs – A Qualitative Analysis”.  The manuscript describes the development of methods for identifying a variety of analytes in different tissues.

Major comments:

  1. Many samples were identified as [M+H–H2O]+ ion, were attempts made to limit the loss of water?
  2. It is unclear what concentration was used for the BACSMLS library for standards – final spiked concentration should be documented.
  3. Line 105: The authors mention that the method has been validated, but it is not clear what “validation” means in this context.  Were FDA and or European guidelines followed?  This should be clarified.
  4. There are numerous methods describing the quantiation of these compounds – the rationale for not include quantitation should be described.

Reviewer 2 Report

The authors present a clear and thorough LC-MS/MS characterization of sterols, bile acids and acylcarnitines using a set of reference standards. They present sufficiently detailed chromatographic and MS/MS data for these groups of compounds. They then apply their method for profiling of the targeted compounds in various biological samples. This is an interesting application that showcase the method but the profiling results are of limited value, since the number of samples of each type is too small (only two of each type) to make any statistical conclusions. Perhaps it would have been more interesting to do a more detailed comparison using fewer types of samples. However, the manuscript is of value for the metabolomics community and contribute useful and detailed data on the studied compounds. I recommend the manuscript to be accepted for publication in Metabolites. The manuscript is well-written with functional English but I would recommend a careful re-read, especially considering the ‘Materials and Methods’ section, with some edits of the language prior to submitting the final version.

Round 2

Reviewer 3 Report

The authors have fully accomplished my requests thus improving the quality of their work.

The paper can be now accepted in the current form.